# DOMAIN ADAPTATION FOR LARGE-VOCABULARY OBJECT DETECTORS

## ABSTRACT

Large-vocabulary object detectors (LVDs) aim to detect objects of many categories, which learn super objectness features and can locate objects accurately while applied to various downstream data. However, LVDs often struggle in recognizing the located objects due to domain **discrepancy** in data distribution and object vocabulary. At the other end, recent vision-language foundation models such as CLIP demonstrate superior open-vocabulary recognition capability. This paper presents KGD, a Knowledge Graph Distillation technique that exploits the implicit knowledge graphs (KG) in CLIP for effectively adapting LVDs to various downstream domains. KGD consists of two consecutive stages: 1) KG extraction that employs CLIP to encode downstream domain data as nodes and their feature distances as edges, constructing KG that inherits the rich semantic relations in CLIP explicitly; and 2) KG encapsulation that transfers the extracted KG into LVDs to enable accurate cross-domain object classification. In addition, KGD can extract both visual and textual KG independently, providing complementary vision and language knowledge for object localization and object classification in detection tasks over various downstream domains. Experiments over multiple widely adopted detection benchmarks show that KGD outperforms the state-of-the-art consistently by large margins.

## 1 INTRODUCTION

Object detection aims to locate and classify objects in images, which conveys critical information about "what and where objects are" in scenes. It is very important in various visual perception tasks in autonomous driving, visual surveillance, object tracking, etc. Unlike traditional object detection, large-vocabulary object detection Li et al. (2022b); Yao et al. (2022); Zhou et al. (2022) aims to detect objects of a much larger number of categories, e.g., 20k object categories in Zhou et al. (2022). It has achieved very impressive progress recently thanks to the availability of large-scale training data. On the other hand, large-vocabulary object detectors (LVDs) often struggle while applied to various downstream tasks as their training data often have different distributions and vocabularies as compared with the downstream data, i.e., due to domain discrepancies.

In this work, we study unsupervised domain adaptation of LVDs, i.e., how to adapt LVDs various downstream tasks with abundant unlabelled data available. Specifically, we observe that LVDs learn superb generalizable objectness knowledge from massive object boxes, being able to locate objects in various downstream images accurately Zhou et al. (2022). However, LVDs often fail to classify the located object due to two major factors: 1) the classic dataset-specific class-imbalance and the resultant distribution bias across domains; and 2) different vocabularies across domains Oksuz et al. (2020); You et al. (2019). At the other end, vision-language models (VLMs) Zhang et al. (2023) such as CLIP Radford et al. (2021) learn from web-scale images and text of arbitrary categories, which achieve significant generalization performance in various downstream tasks with severe domain shifts. Hence, effective adaptation of LVDs towards various unlabelled downstream domains could be facilitated by combining the superior object localization capability from LVDs and the super-rich object classification knowledge from CLIP.

We design Knowledge Graph Distillation (KGD) that explicitly retrieves the classification knowledge of CLIP to adapt LVDs while handling various unlabelled downstream domains. KGD works with one underling hypothesis, i.e., the generalizable classification ability of CLIP largely comes

from its comprehensive knowledge graph learnt over billions of image-text pairs, which enables it to classify objects of various categories accurately. In addition, the knowledge graph in CLIP is implicitly encoded in its learnt parameters which can be exploited in two steps: 1) Knowledge Graph Extraction (KGExtract) that employs CLIP to encode downstream data as nodes and computes their feature distances as edges, constructing an explicit CLIP knowledge graph that captures inherent semantic relations as learnt from web-scale image-text pairs; and 2) Knowledge Graph Encapsulation (KGEncap) that encapsulates the extracted knowledge graph into object detectors to enable accurate object classification by leveraging relevant nodes in the CLIP knowledge graph.

The proposed KGD allow multi-modal knowledge distillation including Language Knowledge Graph Distillation (KGD-L) and Vision Knowledge Graph Distillation (KDG-V). Specifically, KGD-L considers texts as nodes and the distances among text embeddings as edges, enabling detectors to reason whether a visual object matches a text by leveraging other relevant text nodes. KGD-V takes a category of images as a node and the distances among image embeddings as edges, which enhances detection by conditioning on other related visual nodes. Hence, KGD-L and KGD-V complement each other by providing orthogonal knowledge from language and vision perspectives. In this way, KGD allows to explicitly distill generalizable knowledge from CLIP to facilitate unsupervised adaptation of large-vocabulary object detectors towards distinctive downstream datasets.

In summary, the major contributions of this work are threefold. *First*, we propose a knowledge transfer framework that exploits CLIP for effective adaptation of large-vocabulary object detectors towards various unlabelled downstream data. To the best of our knowledge, this is the first work that studies distilling CLIP knowledge graphs for the object detection task. *Second*, we design novel knowledge graph distillation techniques that extracts visual and textual knowledge graphs from CLIP and encapsulates them into object detection networks successfully. *Third*, extensive experiments show that KGD outperforms the state-of-the-art consistently across 10 widely studied detection datasets.

## 2 RELATED WORKS

**Large-vocabulary Object Detection** Dave et al. (2021); Gupta et al. (2019); Redmon & Farhadi (2017); Yang et al. (2019b) aims to detect objects of thousands of classes. Most existing studies tackle this challenge by designing various class-balanced loss functions Dave et al. (2021) for effective learning from large-vocabulary training data and handling the long-tail distribution problem Li et al. (2020); Feng et al. (2021); Wu et al. (2020); Zhang et al. (2021b). Specifically, several losses have been proposed, such as Equalization losses Tan et al. (2020; 2021), SeeSaw loss Wang et al. (2021), and Federated loss Zhou et al. (2021). On the other hand, Yang et al. (2019a) and Detic Zhou et al. (2022) attempt to introduce additional image-level datasets with large-scale fine-grained classes for training large-vocabulary object detector (LVD), aiming to expand the detector vocabulary to tens of thousands of categories. These LVDs learn superb generalizable objectness knowledge from object boxes of massive categories and are able to locate objects in various downstream images accurately Zhou et al. (2022). However, they often fail to classify the located objects Oksuz et al. (2020); You et al. (2019) accurately. In this work, we focus on adapting LVDs towards various unlabelled downstream data by utilizing the super-rich object classification knowledge from CLIP.

**Domain Adaptation** aims to adapt source-trained models towards various target domains. Previous work largely focuses on unsupervised domain adaptation (UDA), which minimizes the domain discrepancy by discrepancy minimization Long et al. (2015); Vu et al. (2019), adversarial training Gong et al. (2019); Vu et al. (2019); Luo et al. (2021), or self-training Lee (2013); Zhang et al. (2019; 2021a). Recently, source-free domain adaptation (SFDA) generates pseudo labels for target data without accessing source data, which performs domain adaptation with entropy minimization Liang et al. (2020), self training Tarvainen & Valpola (2017); Li et al. (2021), contrastive learning Huang et al. (2021); VS et al. (2022), etc. However, most existing domain adaptation methods struggle while adapting LVDs toward downstream domains, largely due to the low-quality pseudo labels resulting from the discrepancy in both data distributions and object vocabulary.

**Vision-Language Models (VLMs)** have achieved great success in various vision tasks Zhang et al. (2023). They are usually pretrained on web-crawled text-image pairs with a contrastive learning objective. Representative methods such as CLIP Radford et al. (2021) and ALIGN Jia et al. (2021) have demonstrated very impressive generalization performance in many downstream vision tasks.

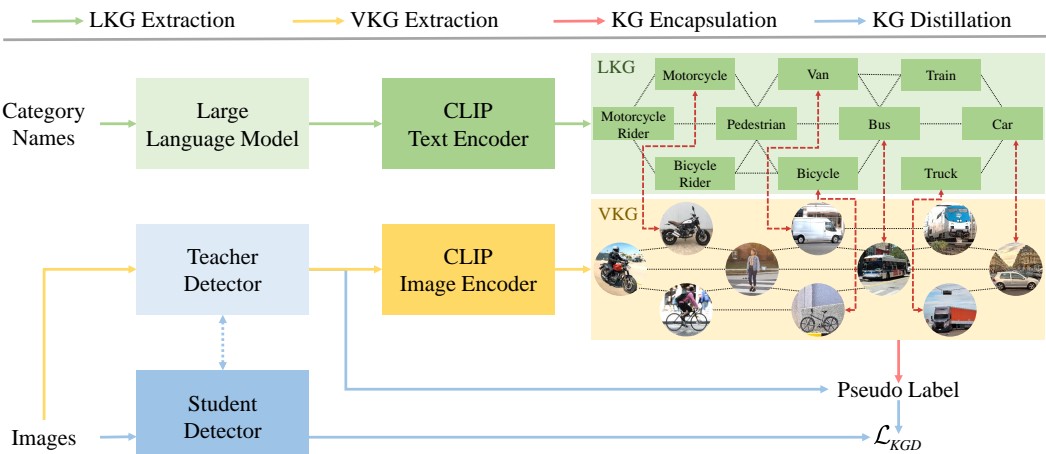

Figure 1: Overview of the proposed Knowledge Graph Distillation (KGD). KGD comprises two consecutive stages including Knowledge Graph Extraction (KGExtract) and Knowledge Graph Encapsulation (KGEncap). KGExtract employs CLIP to encode downstream data as nodes and considers their feature distances as edges, explicitly constructing KGs that inherit the rich semantic relations in CLIP. KGEncap transfers the extracted KGs into the large-vocabulary object detector to enable accurate object classification over downstream data. Besides, KGD works for both image and text data and allow extracting and transferring vision KG and language KG, providing complementary knowledge for adapting large-vocabulary object detectors for handling various unlabelled downstream domains.

Following Radford et al. (2021); Jia et al. (2021), several studies Jia et al. (2021); Kim et al. (2021); Yao et al. (2021); Li et al. (2022a) incorporate cross-attention layers and self-supervised objectives for better cross-modality modelling of noisy data. In addition, several studies Fürst et al. (2022); Doveh et al. (2022); Pei et al. (2022); Gao et al. (2022) learn fine-grained and structural alignment and relations between image and text. In this work, we aim to leverage the generalizable knowledge learnt by VLMs to help adapt LVDs while handling various unlabelled downstream data. However, the aforementioned methods exhibit limited performance when dealing with the downstream domains with

**Knowledge Graph (KG)** Peng et al. (2023) is a semantic network that considers real-world entities or concepts as nodes and treats the semantic relations among them as edges. Multi-modal knowledge graph Alberts et al. (2020); Zhu et al. (2022) extends knowledge from text to the visual domain, enhancing machines' ability to describe and comprehend the real world. These KGs have proven great effectiveness in storing and representing factual knowledge, leading to successful applications in various fields such as entity recognition Zhang et al. (2018); Wilcke et al. (2020), question-answering Marino et al. (2021), and information retrieval Deng et al. (2021). Different from the aforementioned KGs and MMKGs that are often handcrafted by domain experts, we design knowledge graph distillation that builds a LKG and a VKG by explicitly retrieving VLM's generalizable knowledge learnt from web-scale image-text pairs, which effectively uncover the semantic relations across various textual and visual concepts in different downstream tasks, ultimately benefiting the adaptation of LVDs.

## 3 METHOD

**Task Definition.** This paper focuses on unsupervised adaptation of large-vocabulary object detectors (LVDs). We are provided with a set of unlabeled downstream domain data $\mathcal{D}_t = \{\mathbf{x}_i^t\}_{i=1}^{N_t}$ and an LVD pre-trained on labeled source domain detection dataset $\mathcal{D}_s = \{\mathbf{x}_i^s, \mathbf{y}_i^s\}_{i=1}^{N_s}$. $\mathbf{x}_i$ and $\mathbf{y}_i = \{(\mathbf{p}_j, \mathbf{t}_j)\}_{j=1}^{M}$ are the image and $M$ instance annotations of $i$-th sample, where $\mathbf{p}_j$ and $\mathbf{t}_j$ denote the ground-truth category and box coordinate of $j$-th instance. $N_s$ and $N_t$ refer to the number of

samples in $\mathcal{D}_s$ and $\mathcal{D}_t$. The goal is to adapt the pretrained LVD towards the downstream domain $\mathcal{D}_t$ by using the unlabelled images.

**Naïve Solution with Mean Teacher Method (MT) Tarvainen & Valpola (2017).** In this paper, we adopt Detic Zhou et al. (2022) as the pretrained LVD, which utilizes CLIP text embeddings as the classifier. We employ mean teacher Tarvainen & Valpola (2017) as the preliminary solution, which involves a teacher detector and a student detector where the former generates pseudo labels to train the latter while the latter updates the former in a momentum manner. Given a batch of $B$ unlabeled target samples, the teacher detector $\Phi_t$ first produces detection predictions on them, which are then filtered with a predefined threshold $\tau$ to generate detection pseudo label $\hat{\mathbf{y}}_i$ (consisting of classes and bounding boxes). With $\hat{\mathbf{y}}_i$, the unsupervised training of student detector $\Phi_s$ on the unlabeled downstream data can be formulated as the following:

$$Loss = \frac{1}{B}\sum_{i=1}^{B}\mathcal{L}\left(\Phi_s(\mathbf{x}_i^t), \hat{\mathbf{y}}_i\right),\tag{1}$$

where $\mathcal{L}(\cdot) = \mathcal{L}_{rpn}(\cdot) + \mathcal{L}_{reg}(\cdot) + \mathcal{L}_{cls}(\cdot)$ is the detection loss function in which $\mathcal{L}_{rpn}(\cdot)$, $\mathcal{L}_{reg}(\cdot)$, and $\mathcal{L}_{cls}(\cdot)$ denote the loss for region proposal network, regression, and classification, respectively. Note both teacher detector $\Phi_t$ and student detector $\Phi_s$ are initialized with the pretrained LVD.

**Motivation.** On the other hand, although the LVD is able to locate objects in various downstream-domain images accurately Zhou et al. (2022), it often fails to classify the located objects, leading to very noisy detection pseudo labels when serving as the teacher detector. At the other end, vision-language models (VLMs) Zhang et al. (2023) such as CLIP Radford et al. (2021) learns from web-scale images-text pairs of arbitrary categories, which possesses the ability to classify objects accurately in various downstream data. Thus, we argue that effective adaptation of LVDs towards various unlabelled downstream data could be facilitated by combining the superior object localization capability from LVDs and the super-rich object classification knowledge from CLIP. To this end, we design Knowledge Graph Distillation (KGD) with Language KGD and Vision KGD, aiming to explicitly retrieves the classification knowledge of CLIP to adapt LVDs while handling various unlabelled downstream data. The overview of our proposed KGD is shown in Figure 1.

### 3.1 LANGUAGE KNOWLEDGE GRAPH DISTILLATION

The proposed language knowledge graph distillation (KGD-L) aims on distilling knowledge graph from the perspective of text modality. KGD-L works in a two-step manner. The first step is Language Knowledge Graph (LKG) Extraction with Large Language Model Zhao et al. (2023); Ye et al. (2023); Brown et al. (2020) that aims to uncover the implicitly encoded language knowledge in CLIP. With the guidance from the LLM that stores a wide range of knowledge sources from the Internet, LKG Extraction builds a category-discriminative and domain-generalizable LKG. The second step is LKG encapsulation that encapsulates the extracted LKG into the teacher detector, enabling the detector to reason whether a visual object matches a text by leveraging other relevant text nodes and ultimately generate more accurate detection pseudo labels.

**LKG Extraction with LLM.** We first generate domain-generalizable prompts for each object category by leveraging a pretrained LLM. Specifically, given the category set $\mathcal{C} = \{\mathbf{c}_i | i = 1 \ldots, N_c\}$ of a downstream domain , we first obtain the WordNet Miller (1995) Synset definition of category $\mathbf{c}_i$ as follows:

$$\mathbf{d}_i = \text{WNRetrieve}(\mathbf{c}_i),\tag{2}$$

where $\text{WNRetrieve}(\cdot)$ retrieves the WordNet database Miller (1995) and returns the definition of its input. In this way, a category name $\mathbf{c}_i$ can be better defined and described with the informative yet accurate category definition from WordNet. Given a category name $\mathbf{c}_i$ and its definition $\mathbf{d}_i$, we query the LLM with the following prompt template:

*Generate brief descriptions for the appearance of* $[m]$ *types of* $[\mathbf{c}_i]$, $[\mathbf{d}_i]$, *in the context of* $[context]$.

where $[context]$ is a dataset description phrase (e.g., street scenes for Cityscapes Dataset) and $m$ is fixed as 5 for all categories. With the given prompt template, we let LLM generate $m$ domain-generalizable prompts for category $\mathbf{c}_i$ conditioned on its WordNet Synset definition and context:

$$\mathcal{S}_i = \{\mathbf{s}_j\}_{j=1}^{m} = \text{LLM}(\mathbf{c}_i, \mathbf{d}_i, m, context),\tag{3}$$

which is then combined with $\mathbf{d}_i$ as a set of domain generalizable prompts for category $\mathbf{c}_i$:

$$\tilde{\mathcal{S}}_i = \mathcal{S}_i \cup \{\mathbf{d}_i\}, \tag{4}$$

and the domain generalizable prompt set of category set $\mathcal{C}$ can be constructed as the following:

$$\tilde{\mathcal{S}} = \bigcup_{i=1}^{N_c} \tilde{\mathcal{S}}_i. \tag{5}$$

With the category-discriminative and domain-generalizable information contained in $\tilde{\mathcal{S}}$, we formulate the proposed LKG as a weighted undirected graph $G_L = (V_L, U_L)$, which is capable of capturing semantic relationships and associations between different category concepts. $V_L = \{\tilde{\mathbf{s}}_i\}_{i=1}^{N_c(m+1)}$ is the vertex set in which each node $\tilde{\mathbf{s}}_i$ refers to a description in $\tilde{\mathcal{S}}$. And $U_L = \{u_{ij}\}$ is the edge set where each edge $u_{ij}$ denotes the feature cosine similarity between the nodes $\tilde{\mathbf{s}}_i$ and $\tilde{\mathbf{s}}_j$:

$$u_{ij} = \cos \langle T(\tilde{\mathbf{s}}_i), T(\tilde{\mathbf{s}}_j) \rangle = \frac{\delta \cdot T(\tilde{\mathbf{s}}_i)^T T(\tilde{\mathbf{s}}_j)}{||T(\tilde{\mathbf{s}}_i)|| \cdot ||T(\tilde{\mathbf{s}}_j)||} \tag{6}$$

where $T(\cdot)$ refers to the text encoder of CLIP, and $\delta$ is the temperature parameter set as Radford et al. (2021).

**LKG Encapsulation** encapsulates the comprehensive knowledge in the extracted LKG into the teacher detector to facilitate detection pseudo label generation. Specifically, we first employ CLIP to encode the regions cropped by the teacher detector and then generate pseudo labels for each region feature conditioned on LKG. Given the image $\mathbf{x}^t \in \mathcal{D}_t$, we feed it into the teacher detector $\Phi_t$ to acquire the prediction as the following:

$$\hat{\mathbf{y}} = \Phi_t(\mathbf{x}^t), \tag{7}$$

where $\hat{\mathbf{y}} = \{(\hat{\mathbf{p}}_j, \hat{\mathbf{t}}_j)\}_{j=1}^M$, $\hat{\mathbf{p}}_j$ denotes the probability vector of the predicted bounding box $\hat{\mathbf{t}}_j$ after Softmax activation function. $M$ denotes the number of predicted proposals after the thresholding with $\tau$, i.e., a predicted proposal will be discarded if its confidence score is less than $\tau$.

Next, we employ CLIP to encode the predicted object proposals in $\hat{\mathbf{y}}$ as follows:

$$F = V\left(Crop\left(\mathbf{x}^t, \hat{\mathbf{y}}\right)\right), \tag{8}$$

where $Crop(\cdot)$ crops square regions from image $\mathbf{x}^t$ based on the longer edges of bounding boxes in $\hat{\mathbf{y}}$, $V(\cdot)$ is the image encoder of CLIP, and the $j$-th column vector $\mathbf{f}_j$ of matrix $F$ is the feature of $j$-th proposal in $\hat{\mathbf{y}}$.

With the extracted LKG $G_L$ and the features of objects (or object proposals) $F$, we encapsulate the extracted LKG into $\Phi_t$ by reasoning the class of each object conditioned on $G_L$ as follows:

$$\mathbf{p}_{ji}^l = \hat{\mathbf{p}}_{ji} \cdot N\left(\cos \langle \mathbf{f}_j, T(\mathbf{c}_i) \rangle + \max_{\mathbf{s}_k \in \mathcal{S}_i} \left(\cos \langle \mathbf{f}_j, T(\mathbf{s}_k) \rangle\right)\right), \tag{9}$$

where $N(\cdot)$ refers to normalize data to range $[0, 1]$. $\hat{\mathbf{p}}_{ji}$ is the $i$-th element in probability vector $\hat{\mathbf{p}}_j$, which denotes the predicted category probability of $\mathbf{c}_i$. The first term in Eq. 9 denotes the original prediction probability from the teacher model while the second term in Eq. 9 stands for the prediction probability from LKG. $\mathbf{p}_{ji}^l$ denotes the prediction probability calibrated by LKG.

In this way, KGD-L extracts and encapsulates LKG from CLIP into the teacher detector, enabling it to reason whether an object matches a category conditioned on the relevant nodes in LKG and ultimately refining the original detection pseudo labels.

## 3.2 VISION KNOWLEDGE GRAPH DISTILLATION

As LKG captures language knowledge only, we further design vision knowledge graph distillation (KGD-V) that extracts a vision knowledge graph (VKG) and encapsulates it into the teacher detector to improve pseudo label generation. Specifically, VKG captures vision knowledge dynamically along the training process, which complement LKG by providing orthogonal and update-to-date vision information.

**Dynamic VKG Extraction.** We first initialize VKG with the CLIP text embedding and then employ the update-to-date object features to update it using manifold smoothing. Specifically, we initialize VKG as a weighted undirected graph $G_V = (V_V, U_V)$, in which each node $\mathbf{v}_i \in V_V$ is initialized with the CLIP text embedding of category $\mathbf{c}_i$:

$$\mathbf{v}_i = T(\mathbf{c}_i), \tag{10}$$

and the graph edge $u_{ij} \in U_V$ is defined as the cosine similarity between nodes $\mathbf{v}_i$ and $\mathbf{v}_j$. Given a batch of $\{\mathbf{x}_b^t\}_{b=1}^B \subseteq \mathcal{D}_t$ and the corresponding pseudo labels $\{\hat{\mathbf{y}}_b\}_{b=1}^B$ and CLIP features $\{F_b\}_{b=1}^B$, the visual embedding centroid of category $\mathbf{c}_k$ can be obtained as the following:

$$\boldsymbol{\theta}_i = \frac{\sum_{b=1}^B \sum_{\mathbf{f}_j \in \mathbf{F}_b} \mathbf{f}_j \cdot \mathbb{I}(\hat{\mathbf{p}}_j(i) == \hat{\mathbf{p}}_j^{max})}{\sum_{b=1}^B \sum_{\mathbf{f}_j \in \mathbf{F}_b} \mathbb{I}(\hat{\mathbf{p}}_j(i) == \hat{\mathbf{p}}_j^{max})}, \tag{11}$$

where $\hat{\mathbf{p}}_j^{max}$ is the maximum element in probability vector $\hat{\mathbf{p}}_j$, $\mathbb{I}$ is the indicator function. And an affinity matrix $A$ can be calculated as $A_{ij} = exp(-r_{ij}^2/\sigma^2)$ and $A_{ii} = 0$, where $r_{ij} = ||\boldsymbol{\theta}_i - \boldsymbol{\theta}_j||_2$ and $\sigma^2 = \text{Var}(r_{ij}^2)$. In each iteration, the node of VKG is preliminarily updated as:

$$\mathbf{v}_i \leftarrow \lambda \mathbf{v}_i + (1 - \lambda)\boldsymbol{\theta}_i. \tag{12}$$

In order to incorporate the downstream visual graph knowledge into VKG, we perform additional steps to smooth the node of VKG, using the affinity matrix $A$ from the current batch as a guide:

$$\mathbf{v}_i = \sum_j W_{ij}\mathbf{v}_j, \tag{13}$$

where $W = (I - \alpha L)^{-1}$, $L = D^{-\frac{1}{2}}AD^{-\frac{1}{2}}$, $D_{ii} = \sum_j A_{ij}$, $\alpha$ is a scaling factor set as Velazquez et al. (2022), and $I$ is the identity matrix.

**VKG Encapsulation** encapsulate the orthogonal and update-to-date vision knowledge in the extracted VKG into the teacher detector, which complements LKG and further improves pseudo label generation. With the extracted dynamic VKG $G_V$ and the object features $F$ in image $\mathbf{x}^t$, we encapsulate the extracted VKG into $\Phi_t$ in a similar way as the LKG Encapsulation as follows:

$$\mathbf{p}_{ji}^v = \hat{\mathbf{p}}_{ji} \cdot \frac{exp(cos\langle \mathbf{f}_j, \mathbf{v}_i \rangle)}{\sum_{i'} exp(cos\langle \mathbf{f}_j, \mathbf{v}_{i'} \rangle)}, \tag{14}$$

where $\hat{\mathbf{p}}_{ji}$ is the $i$-th element in vector $\hat{\mathbf{p}}_j$, denoting the predicted probability of category $\mathbf{c}_i$. The first term in Eq. 14 is the prediction probability from the teacher model while the second term in Eq. 14 is the prediction probability from VKG. $\mathbf{p}_{ji}^v$ is the prediction probability calibrated by VKG.

In this way, KGD-V extracts and encapsulates the VKG from CLIP into the teacher detector, further refining the detection pseudo labels of visual objects by conditioning on related visual nodes in VKG.

### 3.3 OVERALL OBJECTIVE

Finally, with the pseudo labels $\mathbf{p}_j^l$ and $\mathbf{p}_j^v$ generated from KGD-L and KGD-V respectively, the unsupervised training loss of KGD can be formulated as the following:

$$\mathcal{L}_{KGD} = \sum_{\mathbf{x}^t \in \mathcal{D}_t} \mathcal{L}(\Phi_t(\mathbf{x}^t), \tilde{\mathbf{y}}), \tag{15}$$

where $\tilde{\mathbf{y}} = \{(\tilde{\mathbf{p}}_j, \hat{\mathbf{t}}_j)\}_{j=1}^M$, and $\tilde{\mathbf{p}}_j = N(\mathbf{p}_j^l + \mathbf{p}_j^v)$.

## 4 EXPERIMENTS

This section presents experimental results. Sections 4.1 and 4.2 describe the dataset and implementation details. Section 4.3 presents the experiments across various downstream domain datasets. Section 4.4 and Section 4.5 provide ablation studies and discuss different features of KGD.

Table 1: Benchmarking over autonomous driving datasets under various weather and time conditions. † signifies that the methods employ LLM to generate category descriptions given category names, and CLIP to predict classification pseudo labels for objects. We adopt AP50 in evaluations.

| Method | Cityscapes | Vistas | BDD100K-weather | | | | | BDD100K-time-of-day | | |
|---|---|---|---|---|---|---|---|---|---|---|
| | | | rainy | snowy | overcast | cloudy | foggy | daytime | dawn&dusk | night |
| Detic (Source only) | 46.5 | 35.0 | 34.3 | 33.5 | 39.1 | 42.0 | 28.4 | 39.2 | 35.3 | 28.5 |
| MT | 49.1 | 35.7 | 34.3 | 34.2 | 39.9 | 41.7 | 28.9 | 40.0 | 36.3 | 28.5 |
| MT† | 50.0 | 36.6 | 35.0 | 35.3 | 40.9 | 43.0 | 29.8 | 42.1 | 38.4 | 29.1 |
| SHOT | 49.9 | 36.5 | 34.9 | 34.5 | 40.2 | 42.0 | 34.7 | 40.5 | 36.1 | 26.7 |
| SHOT† | 50.8 | 37.4 | 36.1 | 35.7 | 41.8 | 44.1 | 35.6 | 42.4 | 38.1 | 28.0 |
| SFOD | 49.3 | 35.6 | 32.5 | 33.0 | 40.5 | 43.3 | 33.8 | 40.8 | 36.0 | 28.9 |
| SFOD† | 50.3 | 36.6 | 33.6 | 33.8 | 42.8 | 45.6 | 34.7 | 43.4 | 37.9 | 30.1 |
| HCL | 49.5 | 36.0 | 34.7 | 34.5 | 40.4 | 42.2 | 30.8 | 40.6 | 36.7 | 28.2 |
| HCL† | 50.7 | 37.0 | 35.6 | 35.7 | 42.2 | 44.3 | 31.9 | 42.9 | 38.6 | 29.5 |
| IRG-SFDA | 50.6 | 36.4 | 35.0 | 35.3 | 40.7 | 42.6 | 36.4 | 40.8 | 36.4 | 27.8 |
| IRG-SFDA† | 51.7 | 37.5 | 35.9 | 36.4 | 42.6 | 44.8 | 36.7 | 43.0 | 38.3 | 28.9 |
| **KGD (Ours)** | **53.6** | **40.3** | **37.3** | **37.1** | **44.6** | **48.2** | **38.0** | **46.6** | **41.0** | **31.2** |

## 4.1 DATASETS

We perform experiments on 11 object detection datasets that span different downstream domains including the object detection for autonomous driving Cordts et al. (2016); Neuhold et al. (2017), autonomous driving under different weather and time-of-day conditions Yu et al. (2018), intelligent surveillance Luo et al. (2018); Yongqiang et al. (2021); Zhu et al. (2021), common objects Everingham et al. (2015); Shao et al. (2019), and artistic illustration Inoue et al. (2018). More dataset details can be found in the Appendix.

## 4.2 IMPLEMENTATION DETAILS

We adopt Detic Zhou et al. (2022) as LVD, where CenterNet2 Zhou et al. (2021) with Swin-B Liu et al. (2021) is pre-trained on LVIS Gupta et al. (2019) for object localization and ImageNet-21K Deng et al. (2009) for object classification. During adaption, the updating rate of EMA detector is set as 0.9999. The pseudo labels generated by the teacher detector with confidence greater than the threshold $\tau = 0.25$ are selected for adaptation. We use AdamW Loshchilov & Hutter (2017) optimizer with initial learning rate $5 \times 10^{-6}$ and weight decay $10^{-4}$, and adopt a cosine learning rate schedule without warm-up iterations. The batch size is 2 and the image's shorter side is set to 640 while maintaining the aspect ratio unchanged.

## 4.3 RESULTS

Tables 1-3 show the benchmarking of our methods with state-of-the-art domain adaptive detection methods. As there are few prior studies on LVD adaptation, we compare our proposed method with state-of-the-art source-free domain adaptation methods for benchmarking, including Mean Teacher (MT) Tarvainen & Valpola (2017), SHOT Liang et al. (2020), SFOD Li et al. (2021), HCL Huang et al. (2021), and IRG-SFDA VS et al. (2022). For fair comparison, we incorporate CLIP Radford et al. (2021) and LLM Brown et al. (2020) into the compared methods (marked with †). Specifically, we employ LLM Brown et al. (2020) to generate category descriptions given category names, and CLIP Radford et al. (2021) to predict pseudo labels for object classification.

**Object detection for autonomous driving.** As Table 1 shows, the proposed KGD outperforms the baseline substantially over the general autonomous driving datasets Cityscapes and Vistas (with an average improvement of 6.20 in AP50). KGD also outperforms the state-of-the-art by 2.35 on average, demonstrating the superiority of KGD in adapting pretrained LVDs toward autonomous driving scenarios with substantial inter-domain discrepancy. In addition, Table 1 shows experiments on autonomous driving data under various weather and time conditions. We can observe that KGD still achieves superior detection performance even though the unlabeled target data experience large style variation and severe quality degradation. Further, the experiments show that KGD still outperforms the state-of-the-art clearly when CLIP and LLM are incorporated, validating that the performance gain largely comes from our novel knowledge graph distillation instead of merely using CLIP and LLM.

Table 2: Benchmarking over intelligent surveillance datasets. † signifies that the methods employ LLM to generate category descriptions given category names, and CLIP to predict classification pseudo labels for objects. We adopt AP50 in evaluations.

| Method | MIO-TCD | BAAI-VANJEE | VisDrone |
|---|---|---|---|
| Detic (Source only) | 20.6 | 20.6 | 19.0 |
| MT | 20.0 | 23.4 | 18.9 |
| MT† | 20.9 | 23.9 | 20.4 |
| SHOT | 21.2 | 22.5 | 19.4 |
| SHOT† | 22.3 | 23.3 | 20.9 |
| SFOD | 19.8 | 22.8 | 18.8 |
| SFOD† | 21.0 | 23.1 | 20.2 |
| HCL | 20.5 | 23.6 | 18.8 |
| HCL† | 21.1 | 24.1 | 19.6 |
| IRG-SFDA | 20.7 | 22.8 | 18.8 |
| IRG-SFDA† | 21.6 | 23.7 | 20.0 |
| **KGD (Ours)** | **24.6** | **24.3** | **23.7** |

Table 3: Benchmarking over common objects and artistic illustration datasets. † signifies that the methods employ LLM to generate category descriptions given category names, and CLIP to predict classification pseudo labels for objects. We adopt AP50 in evaluations.

| Method | Common objects | | Artistic illustration | | |
|---|---|---|---|---|---|
| | Pascal VOC | Objects365 | Clipart1k | Watercolor2k | Comic2k |
| Detic (Source only) | 83.9 | 29.4 | 61.0 | 58.9 | 51.2 |
| MT | 85.6 | 31.0 | 62.7 | 58.4 | 49.8 |
| MT† | 86.2 | 31.4 | 63.4 | 59.6 | 51.1 |
| SHOT | 84.0 | 30.7 | 61.3 | 58.3 | 50.4 |
| SHOT† | 84.5 | 31.2 | 62.3 | 59.8 | 52.1 |
| SFOD | 85.5 | 31.6 | 63.4 | 58.2 | 50.1 |
| SFOD† | 86.2 | 32.0 | 64.6 | 59.3 | 51.8 |
| HCL | 85.8 | 31.8 | 63.1 | 58.3 | 52.3 |
| HCL† | 86.5 | 32.3 | 64.7 | 59.7 | 53.7 |
| IRG-SFDA | 86.0 | 32.0 | 63.3 | 60.8 | 50.4 |
| IRG-SFDA† | 86.3 | 32.3 | 65.0 | 61.5 | 52.0 |
| **KGD (Ours)** | **86.9** | **34.4** | **69.1** | **63.5** | **55.6** |

**Object detection for intelligent surveillance.** The detection results on intelligent surveillance datasets are presented in Table 2. Notably, the proposed KGD surpasses all other methods by significant margins, which underscores the effectiveness of KGD in adapting the pretrained LVD towards the challenging surveillance scenarios with considerable variations in camera lenses and angles. The performance improvements achieved by KGD in this context demonstrate its effectiveness in exploring the unlabeled surveillance datasets by retrieving the classification knowledge of CLIP.

**Object detection for common objects.** We evaluate the effectiveness of our KGD on the common object detection task using Pascal VOC and Objects365. Table 3 reports the detection results, showcasing significant improvements over the baseline and outperforming state-of-the-arts, thereby highlighting the superiority of KGD. Besides, we can observe that the performance improvements on the Pascal VOC dataset and Objects365 dataset are not as significant as those in autonomous driving. This discrepancy is attributed to the relatively smaller domain gap between common objects and the pretraining dataset of LVD.

**Object detection for artistic illustration.** Table 3 reports the detection results on artistic illustration datasets. The proposed KGD outperforms all other methods by substantial margins, which highlights the effectiveness of KGD in adapting the pretrained large-vocabulary object detector towards artistic images that exhibit distinct domain gaps with natural images.

## 4.4 ABLATION STUDIES

In Table 4, we conducted ablation studies to assess the individual contribution of our proposed KGD-L and KGD-V on the task of LVD adaptation. The pretrained LVD (i.e., Detic Zhou et al. (2022) without adaptation) does not perform well due to the significant variations between its pretraining data and the downstream data, As a comparison, either KGD-L or KGD-V brings significant performance improvements (i.e., +6.3 of AP50 and +6.2 of AP50 over the baseline), demonstrating both language and vision knowledge graphs built from CLIP can clearly facilitate the unsupervised adaptation of large-vocabulary object detectors. The combination of KGD-L and KGD-V performs

Table 4: Ablation studies of KGD with KGD-L and KGD-V. The experiments are conducted on the Cityscapes dataset.

| Method | Language Knowledge Graph Distillation | Vision Knowledge Graph Distillation | AP50 |
|---|---|---|---|
| Detic (Source only) | | | 46.5 |
| **KGD (Ours)** | ✓ | | 52.8 |
| | | ✓ | 52.7 |
| | ✓ | ✓ | **53.6** |

the best clearly, showing that our KGD-L and KGD-V are complementary by providing orthogonal language and vision knowledge for regularizing the unsupervised adaptation of LVDs.

## 4.5 DISCUSSION

**Comparisons with existing CLIP knowledge distillation methods for detection.** We compared our KGD with existing CLIP knowledge distillation methods designed for detection tasks. Most existing methods achieve CLIP knowledge distillation by mimicking its feature space, such as VILD Gu et al. (2021), RegionKD Rasheed et al. (2022), and OADP Wang et al. (2023). Table 5 reports the experimental results over the Cityscapes dataset, which shows existing CLIP knowledge distillation methods do not perform well in adapting LVDs to downstream tasks. The main reason is that they merely align the feature space between LVDs and CLIP without considering the inherent semantic relationships between different object categories. KGD also performs knowledge distillation but works for LVDs adaption effectively, largely because it works by extracting and encapsulating knowledge CLIP knowledge graphs which enables accurate object classification by leveraging relevant nodes in the knowledge graphs.

Table 5: Comparisons with existing CLIP knowledge distillation methods on LVD adaptation. For a fair comparison, we incorporate them with Mean Teacher Method (the columns with 'MT+').

| Method | Detic (Source only) | MT | MT+VILD | MT+RegionKD | MT+OADP | **KGD (Ours)** |
|---|---|---|---|---|---|---|
| AP50 | 46.5 | 49.1 | 50.6 | 50.2 | 50.2 | **53.6** |

**Qualitative experimental results.** We present qualitative results of KGD over diverse downstream domain detection datasets in the Appendix. The qualitative results illustrate the effectiveness of KGD in producing accurate detection results across various domains, thereby qualitatively demonstrating its capability to adapt LVDs to unlabelled downstream domains with significant discrepancy in data distribution and vocabulary.

**Parameter studies.** In the pseudo label generation in KGD, the reliable pseudo labels are acquired with a pre-defined confidence threshold $\tau$. We studied $\tau$ by changing it from $0.15$ to $0.35$ with a step of $0.05$. Table 6 reports the experiments over the Cityscapes dataset. It shows that $\tau$ does not affect KGD clearly, demonstrating the proposed KGD is tolerant to hyper-parameters.

Table 6: Parameter analysis of KGD for the pseudo label generation threshold $\tau$.

| $\tau$ | 0.15 | 0.2 | 0.25 | 0.3 | 0.35 |
|---|---|---|---|---|---|
| AP50 | 53.4 | 53.2 | 53.6 | 53.9 | 53.5 |

## 5 CONCLUSION

This paper presents KGD, a novel knowledge distillation technique that exploits the implicit KG of CLIP to adapt large-vocabulary object detectors for handling various unlabelled downstream data. KGD consists of two consecutive stages including KG extraction and KG encapsulation which extract and encapsulate visual and textual KGs simultaneously, thereby providing complementary vision and language knowledge to facilitate unsupervised adaptation of large-vocabulary object detectors towards various downstream detection tasks. Extensive experiments on multiple widely-adopted detection datasets demonstrate that KGD consistently outperforms state-of-the-art techniques by clear margins.

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

# A    APPENDIX

## A.1    DATASETS

**Cityscapes Cordts et al. (2016)** is a dataset designed for the purpose of understanding street scenes. It comprises images captured in 50 different cities, encompassing a total of 2975 training images and 500 validation images. These images are captured under normal weather conditions with pixel-wise instance annotations of 8 categories.

**Vistas Neuhold et al. (2017)** is an autonomous driving dataset collected for street scene understanding. It comprises a vast collection of high-resolution images that encompass diverse urban environments from various locations worldwide. The dataset consists of 18000 training images and 2000 validation images with pixel-wise instance annotations.

**BDD100k Yu et al. (2018)** is a large-scale driving video dataset with a wide range of diverse driving scenarios. It comprises various weather conditions such as clear, cloudy, overcast, rainy, snowy, and foggy, as well as different times of the day including dawn, daytime, and night. The dataset contains 70000 training videos and 10000 validation videos. Each video is annotated with bounding boxes for objects of 10 distinct categories.

**MIO-TCD Luo et al. (2018)** is an intelligent surveillance dataset collected for traffic analysis. It comprises 137743 images captured at different times of the day and various periods throughout the year. The images are captured from diverse viewing angles. Each image in the dataset is annotated with bounding boxes, providing precise spatial locations of objects of 11 categories.

**BAAI-VANJEE Yongqiang et al. (2021)** is a dataset collected for surveillance applications. It comprises 5000 high-quality images captured by the VANJEE smart base station positioned at a height of 4.5 meters. Each image in the dataset is annotated with bounding boxes, providing spatial locations of objects of 12 categories.

**VisDrone Zhu et al. (2021)** is a surveillance dataset captured using drone-mounted cameras in different scenarios, and under various weather and lighting conditions. It comprises 288 video clips with 261908 frames, as well as an additional set of 10209 static images. These frames and images are annotated with more than 2.6 million bounding boxes of objects of 10 categories.

**Pascal VOC Everingham et al. (2015)** consists of two distinct sub-datasets: Pascal VOC 2007 and Pascal VOC 2012. The former comprises a total of 2501 training images and 2510 validation images, while the latter encompasses a larger set of 5717 training images and 5823 validation images. This dataset includes bounding box annotations of 20 object categories.

**Objects365 Shao et al. (2019)** is a large-scale object detection dataset with 2 million images, 30 million bounding boxes, and 365 categories, which is designed for detecting diverse objects in the wild.

**Clipart1k Inoue et al. (2018)** is a prominent dataset employed in cross-domain object detection, comprising 1000 clipart images collected from one dataset (CMPlaces Castrejon et al. (2016)) and two image search engines (Openclipart[1] and Pixabay[2]). Each image in the dataset is annotated with bounding boxes for objects that share 20 categories with Pascal VOC Everingham et al. (2015).

**Watercolor2k Inoue et al. (2018)** comprises a collection of 2000 watercolor images with image and instance-level annotations of 6 categories. It is also a prominent dataset employed in cross-domain object detection.

**Comic2k Inoue et al. (2018)** contains 2000 comic images with image and instance-level annotations, sharing 6 categories with Pascal VOC Everingham et al. (2015).

## A.2 ALGORITHM OF KGD

We describe the detailed algorithm of our proposed KGD in Algorithm 1.

---

**Algorithm 1:** Adapting Open-Vocabulary Detectors via Knowledge Graph Distillation

---

**Input:** unlabeled downstream data $\mathcal{D}_t$, pretrained LVD $\Phi$, CLIP image encoder $V$, CLIP text encoder $T$, WordNet database retrieval function WNDefRetrieve;

**Output:** domain adaptive detector $\Phi_s$;

1 Initialization: teacher detector $\Phi_t \leftarrow \Phi$, student detector $\Phi_s \leftarrow \Phi$, maximum iteration $l$, momentum updating frequency $t_{mom}$, momentum updating rate $\mu$;

2 Extract LKG by Eq.(2)-(6);

3 Extract VKG by Eq.(10);

4 **for** $t \leftarrow 0$ **to** $l$ **do**

5      Sample a batch of $B$ targe domain samples: $\{\mathbf{x}_b^t\}_{b=1}^B \subseteq \mathcal{D}_t$;

6      Generate pseudo label set $\{\hat{\mathbf{y}}_b\}_{b=1}^B$ by Eq.(7);

7      Generate CLIP feature matrix set $\{F_b\}_{b=1}^B$ with Eq.(8);

8      Encapsulate LKG by Eq.(9);

9      Encapsulate VKG by Eq.(14);

10      Minimize overall objective function Eq.(15) by updating $\Phi_s$;

11      Update VKG by Eq.(12) and (13);

12      **if** $t \% t_{mom} == 0$ **then**

13          Update EMA detector: $\Phi_t \leftarrow \mu\Phi_t + (1 - \mu)\Phi_s$;

---

## A.3 ADDITIONAL DISCUSSION

**Language Knowledge Graph Distillation Strategies** Our proposed Language Knowledge Graph Distillation(KGD-L) introduces Large Language Model (LLM) Zhao et al. (2023); Ye et al. (2023); Brown et al. (2020) to uncover the implicitly encoded language knowledge in CLIP Radford et al. (2021) and accordingly enables to build a category-discriminative and domain-generalizable Language Knowledge Graph (LKG) as described in Section 3.2 We examine the superiority of the proposed LKG Extraction with LLM by comparing it with "LKG Extraction with category names" and "LKG Extraction with WordNet Miller (1995) Synset definitions", the former builds LKG directly with the category names from downstream datasets while the latter builds LKG using WordNet Synset definitions that are retrieved from the WordNet database with category names from downstream datasets. As Table 7 shows, both strategies achieve sub-optimal performance. For "LKG Extraction with category names", the category names are often ambiguous and less informative which degrades adaptation. For "LKG Extraction with WordNet Synset definitions", the used WordNet Synset definitions are more category-discriminative but they often have knowledge gaps with downstream data, limiting adaptation of the pretrained large-vocabulary detectors (LVDs). As a comparison, our proposed LKG Extraction performs clearly better due to the guidance of LLM that captures super-rich knowledge from the Internet which helps generate category-discriminative and

---

[1]https://openclipart.org/

[2]https://pixabay.com/

domain-generalizable LKG and facilitates the adaption of LOVDs towards downstream data effectively.

Table 7: Study of different Language Knowledge Graph Distillation(KGD-L) strategies. The experiments are conducted on the Cityscapes dataset.

| Method | Language Knowledge Graph Distillation | | | AP50 |
| --- | --- | --- | --- | --- |
| | LKG Extraction with category names | LKG Extraction with WordNet Synset definitions | LKG Extraction with LLM | |
| Detic (Source only) | | | | 46.5 |
| **KGD-L only** | ✓ | | | 51.9 |
| | | ✓ | | 52.0 |
| | | | ✓ | **52.8** |

**Vision Knowledge Graph Distillation Strategies** Our proposed Vision Knowledge Graph Distillation(KGD-L) captures the Dynamic vision knowledge graph (VKG) along the training as described in Section 3.2, which complements LKG by providing orthogonal and update-to-date vision information. We examine the proposed Dynamic VKG Extraction by comparing it with "Static VKG Extraction". "Static VKG Extraction" builds a static VKG with CLIP features of image crops of objects that are predicted by the pretrained LVD before adaptation. It remains unchanged during the LVD adaptation process. As Table 8 shows, "Static VKG Extraction" does not perform well in model adaptation, largely because the extracted static VKG is biased towards the pretraining datasets of the LVD and impedes domain-specific adaptation. As a comparison, our proposed Dynamic VKG Extraction shows clear improvements as the update-to-date vision information extracted along the training process dynamically stabilizes and improves the model adaptation.

Table 8: Studies of different Vision Knowledge Graph Distillation(KGD-V) strategies. The experiments are conducted on the Cityscapes dataset.

| Method | Vision Knowledge Graph Distillation | | AP50 |
| --- | --- | --- | --- |
| | Static VKG Extraction | Dynamic VKG Extraction | |
| Detic (Source only) | | | 46.5 |
| **KGD-V only** | ✓ | | 51.9 |
| | | ✓ | **52.7** |

**Distance metrics for constructing Knowledge Graph.** We explore the feature distance metrics for constructing knowledge graphs. We conduct experiments that construct knowledge graphs with the following feature distance metrics: 1) Cosine Similarity Deza et al. (2009), 2) Euclidean Distance Deza et al. (2009), 3) Manhattan Distances Deza et al. (2009). The results in Table 9 show that our KGD works effectively and consistently with different feature distance metrics. Besides, the cosine similarity metric performs the best, largely because CLIP is also trained with cosine similarity where using the same metric to distill its knowledge works the best reasonably.

Table 9: Study of different distance metrics for constructing KG. The experiments are conducted on the Cityscapes dataset.

| Distance Metrics | Cosine Similarity | Euclidean Distance | Manhattan Distances |
| --- | --- | --- | --- |
| AP50 | 53.6 | 52.9 | 53.1 |

**Analysis of KGD with respect to the parameter $m$.** In Eq. 3, the parameter $m$ controls the number of domain-generalizable prompts generated by LLM for a certain category. It plays a critical role in balancing the trade-off between generalization and discrimination in pseudo label denoising with our KGD. We study $m$ by changing it from 1 to 9 with a step of 2 and the table below shows results on Cityscapes. It shows that the increase of $m$ improves the performance clearly.

**Training and inference time analysis.** We study the training and inference time of all the compared methods, and Table 11 shows the results on Cityscapes. It shows that incorporating CLIP into

Table 10: Parameter analysis of KGD with respect to the parameter $m$.

| $m$ | 1 | 3 | 5 | 7 | 9 |
|---|---|---|---|---|---|
| AP50 | 50.9 | 52.5 | 53.9 | 54.1 | 53.9 |

unsupervised domain adaptation introduces a few additional overhead on training time and almostly does not affect inference time. The reason lies in that the cropped object regions are processed by CLIP in a parallel manner during training while the inference pipeline does not involve CLIP.

Table 11: Training and inference time analysis of all the compared methods. The experiments are conducted on one RTX 2080Ti. † signifies that the methods employ LLM to generate category descriptions given category names, and CLIP to predict classification pseudo labels for objects.

| Method | MT | MT† | SHOT | SHOT† | SFOD | SFOD† | HCL | HCL† | IRG-SFDA | IRG-SFDA† | KGD (Ours) |
|---|---|---|---|---|---|---|---|---|---|---|---|
| Training Time (hours) | 4.083 | 5.022 | 4.055 | 5.045 | 4.110 | 5.193 | 4.133 | 5.095 | 4.158 | 5.222 | 5.267 |
| Inference Speed (images per second) | 6.703 | 6.767 | 6.749 | 6.809 | 6.523 | 6.752 | 6.689 | 6.683 | 6.758 | 6.701 | 6.758 |

**Study of vocabulary size of KGD.** The construction of knowledge graphs greatly improves the quality of pseudo-labels. We study how vocabulary size affects the cost of model training and inference. As shown in Table 12, the increase in vocabulary size of knowledge graphs brings little computation overhead, largely because our knowledge graphs are implemented in a efficient computation manner.

Table 12: Study of vocabulary size of knowledge graphs. The experiments are conducted on one RTX 2080Ti.

| Dataset | Vocabulary Size | Training Time (hours) | Inference Speed (images per second) |
|---|---|---|---|
| Watercolor2k | 6 | 5.165 | 6.696 |
| Comic2k | 6 | 5.159 | 6.721 |
| Cityscapes | 8 | 5.167 | 6.718 |
| Vistas | 8 | 5.163 | 6.694 |
| VisDrone | 10 | 5.169 | 6.720 |
| BDD100k | 10 | 5.168 | 6.788 |
| MIO-TCD | 11 | 5.167 | 6.677 |
| BAAI-VANJEE | 12 | 5.169 | 6.714 |
| Clipart1k | 20 | 5.165 | 6.721 |
| Pascal VOC | 20 | 5.168 | 6.698 |
| Objects365 | 365 | 5.171 | 6.723 |

## A.4 MORE QUALITATIVE COMPARISONS

We provide qualitative illustrations of KGD over downstream datasets.

As shown in Figure 2-6, KGD produces accurate detection across multiple datasets, demonstrating its capability to adapt LVDs to various downstream domains of very different data distribution and vocabulary.

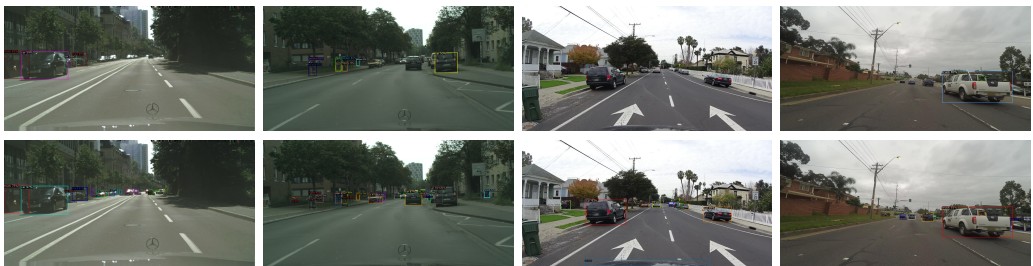

Figure 2: Qualitative comparisons over autonomous-driving data. Zoom in for details. Top: Detic Zhou et al. (2022). Bottom: KGD (Ours).

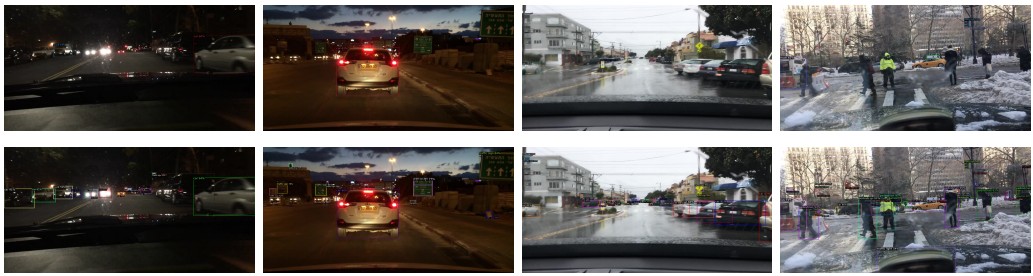

Figure 3: Qualitative comparisons over autonomous-driving data under different weather and time-of-day conditions. Zoom in for details. Top: Detic Zhou et al. (2022). Bottom: KGD (Ours).

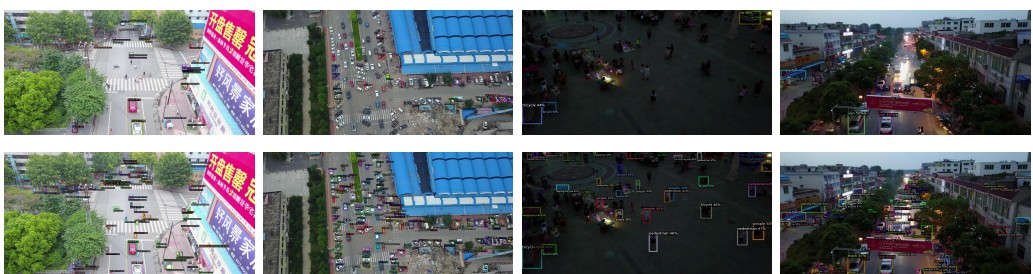

Figure 4: Qualitative comparisons over intelligent-surveillance data. Zoom in for details. Top: Detic Zhou et al. (2022). Bottom: KGD (Ours).

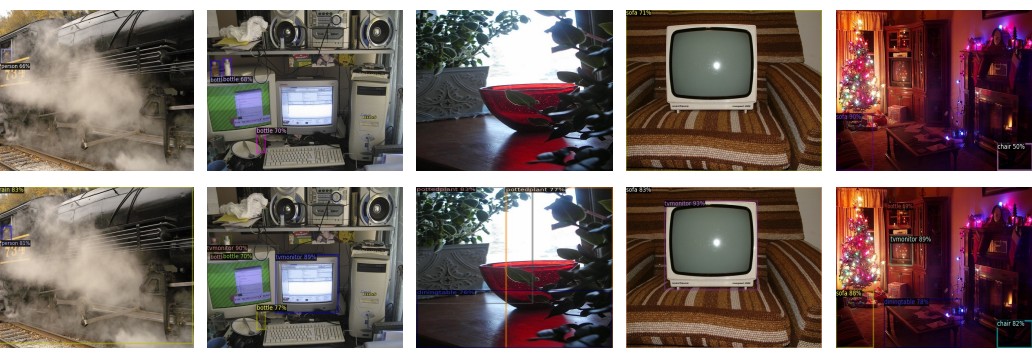

Figure 5: Qualitative comparisons over common-object data. Zoom in for details. Top: Detic Zhou et al. (2022). Bottom: KGD (Ours).

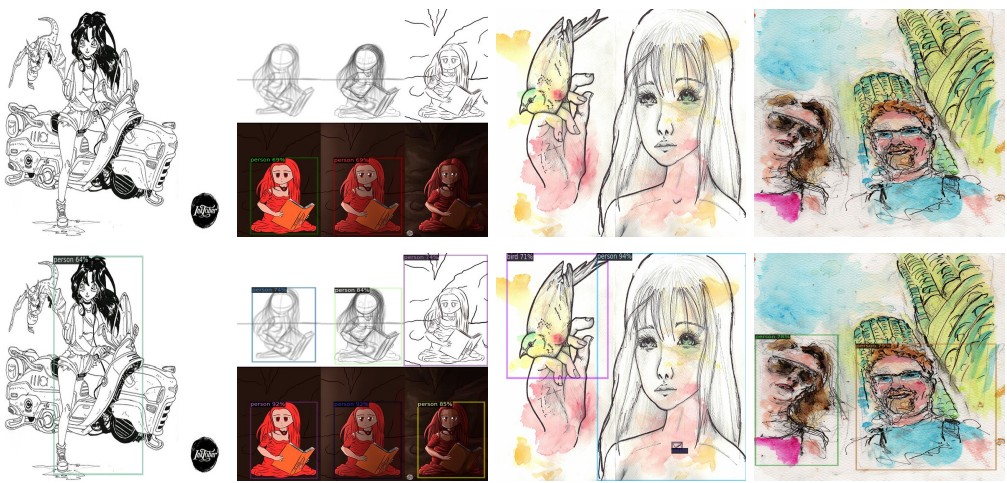

Figure 6: Qualitative comparisons over artistic illustration data. Zoom in for details. Top: Detic Zhou et al. (2022). Bottom: KGD (Ours).

