# OpenReview forum: "Domain Adaptation for Large-Vocabulary Object Detectors"
_ICLR.cc/2024/Conference — ICLR 2024 Conference Withdrawn Submission_

### Official Review · Reviewer_HDW7 · 2023-10-30

**Soundness:** 2 fair
**Presentation:** 2 fair
**Contribution:** 2 fair
**Rating:** 5
**Confidence:** 4

**Summary:**

This paper proposes a distillation method based on knowledge graph extraction and encapsulation to exploit the pretrained CLIP to help a pretrained detectors to generalize onto new domain and vocabularies. The proposed Language Knowledge Graph Distillation (L-KGD) and Vision-KGD collaboratively improve the generalization performance. The resulting detector outperforms the prior works in different datasets including BDD100K, MIO-TCD, VisDrone, Objects365, Clipart datasets.

**Strengths:**

* The adaptation of the detector to in-the-wild data and novel domain is an important problem.

* The proposed method attempts to advance the knowledge graph based distillation to exploit the pretrained CLIP to help adapting detectors to generalize to new vocabularies.

* The proposed method shows improvements over the baseline on different datasets.

**Weaknesses:**

* Unfinished sentence in Related Works section: "However, the aforementioned methods exhibit limited performance when dealing with the downstream domains with"

* Motivation is not supported by experiment or any evidence:  On the other hand, although the LVD is able to locate objects in various downstream domain images accurately Zhou et al. (2022), it often fails to classify the located object

* Missing FoggyCityscapes / Cityscapes benchmark results: IRG-SFDA (VS et al, 2022) reports the adaptation results on “Cityscapes → FoggyCityscapes” and “Sim10K → Cityscapes” benchmarks. It would be more helpful for this paper to have the comparison on the same benchmarks to provide easier comparison to the SOTA methods.
Detector capacity: The IRG-SFDA uses Faster R-CNN based on R-50 backbone, whereas this work uses CenterNet2 detector based on Swin-B backbone which is known to be much stronger. Do all other compared methods in Table 1, 2, and 3 use the same detector?

* Comparison with Rasheed et al. and OADP methods (Sec 4.5, Table 5): These two methods are open-vocabulary detection (OVD) methods which also adopt CLIP distillation for detection. As their main benchmark is COCO-OVD and LVIS-OVD instead of Cityscapes, it is strongly suggested that this work also uses the same benchmarks to compare with the official records of the SOTA works.

**Questions:**

* Do all compared methods in Table 1, 2, 3 use the same detector (CenterNet2) and backbone capacity (Swin-B)?

* How would the proposed method be compared with the FoggyCityscapes benchmark and Open-Vocabulary Detection benchmark?

* Please refer to the Weakness.

---

### Official Review · Reviewer_iHrk · 2023-10-31

**Soundness:** 2 fair
**Presentation:** 3 good
**Contribution:** 2 fair
**Rating:** 5
**Confidence:** 4

**Summary:**

A Knowledge Graph Distillation technique is proposed for Large-vocabulary object detectors, that exploits the implicit knowledge graphs in CLIP to adapt LVDs to downstream domains. A language knowledge graph and a vision knowledge graph are constructed, where the textual and visual knowledges of CLIP are adopted, respectively. The presented Knowledge Graph Distillation transfers these knowledges to detectors by utilizing logits distillation. Experiments conducted on multiple detection benchmarks show good performance.

**Strengths:**

Overall, this paper is easy to read.
The proposed KGD achieves great performance.

**Weaknesses:**

The motivation for using graph is confusion. This paper just uses the independent information of vertices in the graph, while the edges are ignored.

The novelty of this paper is limited. Compared with [1], the main difference is that this paper uses a simple late-fusion strategy to better utilize multimodal knowledge. It seems that this work generates pseudo-labels similar to other CLIP-based methods, except that other CLIP-based methods generate one-hot pseudo labels, while this work produces soft pseudo labels.
[1] Gu, X., Lin, T.Y., Kuo, W. and Cui, Y., 2022. Open-vocabulary object detection via vision and language knowledge distillation. In ICLR.

Experiments demonstrate the performance improvement. However, the reasons caused the improvement is unclear. As listed in Table 10, the gain appears to come mainly from the prior information generated by LLM rather than the proposed KGD. It needs to be shown that other CLIP-based methods cannot achieve such a large gain from the prior information produced by LLM.

The details in Section 3.1 are ambiguous. Whether the [content] in the prompt template is a human-provided prior per dataset (e.g., street scenes for Cityscapes Dataset)? If this is the case, the experiments are unfair.

**Questions:**

See above comments.

---

### Official Review · Reviewer_xetS · 2023-10-31

**Soundness:** 3 good
**Presentation:** 2 fair
**Contribution:** 3 good
**Rating:** 6
**Confidence:** 3

**Summary:**

This paper focuses on solving the "domain discrepancy" problem of unsupervised domain adaptation of large vocabulary object detectors (LVDs). To solve this problem, the authors propose a method called "Knowledge Graph Distillation" (KGD). This approach consists of two primary steps: Knowledge Graph Extraction (KGExtract) and Knowledge Graph Encapsulation (KGEncap).

**Strengths:**

1) This paper presents a valuable and novel insight: To facilitate effective adaptation of LVDs towards various unlabelled downstream data, the authors design Knowledge Graph Distillation (KGD). This approach aims to explicitly retrieve the classification knowledge from CLIP and distill CLIP knowledge graphs, thereby combining the superior object localization capabilities of LVDs with CLIP's super-rich object classification knowledge.

2) Extensive experiments conducted on 11 target detection datasets covering multiple different downstream application fields demonstrate the excellent performance of the method proposed by the author.

3) The structure of this method is clear.

**Weaknesses:**

Some details of the paper are not clear. See questions.

**Questions:**

1) In Eq. 3, the authors state that the parameter m is fixed as 5. However, Table 10 shows that m=7 yields a better result than m=5 on the Cityscapes dataset. Why wasn't m=7 chosen for the experiments? Perhaps the authors could further explain their choice. Is it due to computational complexity, or are there other reasons?

2) In Eq. 6, the specific value of the temperature parameter \delta is not made clear. Additionally, this value is not discussed in either the implementation details or the ablation study. The authors should specify the value of the parameter, or explain how to choose the parameter.

3) Eq.9 and Eq.14 outline methods for calibrating the teacher detector prediction probability using the extracted LKG and VKG. However, in these two equations, I feel it is not clear enough and a little hard to fully understand the use of weighted undirected graph from the "Extraction" step. How is the built weighted undirected graph used in the 'Encapsulation' step?

4) In Eq.13, the specific value of the scaling factor \alpha is not made clear. And I think the authors could provide more explanation about how this additional smoothing operation working. If willing, some experiments could be provided to demonstrate the effect of the smoothing operation.

---

### Official Review · Reviewer_pudg · 2023-11-01

**Soundness:** 3 good
**Presentation:** 4 excellent
**Contribution:** 3 good
**Rating:** 5
**Confidence:** 4

**Summary:**

This paper proposes a Knowledge Graph Distillation (KGD) technique to adapt retrained large-vocabulary object detectors to different downstream domains. The idea is to first extract KG from CLIP and then transfer the extracted KD into detectors. Experiments show that KGD outperforms the baseline methods over multiple benchmarks.

**Strengths:**

+ The writing is clear and easy to follow.

+ The problem of adapting LVDs to downstream domains has practical value.

+ Extensive experiments have been conducted over 11 benchmarks.

**Weaknesses:**

- The proposed KGD technique targets at LVDs. However, there have been plenty of open-vocabulary detectors (OVDs). Do OVDs face the same domain adaptation problem? If so, can KGD be applied to them?

- Detic is pretrained on ImageNet-21K, which may contain the categories from the downstream datasets. The authors are encouraged to provide how many categories of the test datasets appear in ImageNet-21K.

- More ablation studies are expected. For instance, how much benefit is brought by LLM during LKG extraction?

**Questions:**

- Is KGD applicable in training LVDs from scratch? In other words, is KGD still effective if the pretrained LVD model is poorly trained or even randomly initialized?

---

### Official Review · Reviewer_MJjx · 2023-11-06

**Soundness:** 3 good
**Presentation:** 3 good
**Contribution:** 3 good
**Rating:** 5
**Confidence:** 4

**Summary:**

This paper proposes domain adaption methods for large vocabulary objectors, the method is based on constructing and distilling the knowledge graph with a pre-trained multi-modal model. The paper conducts various experiments with different adaption senarios to validate the method.

**Strengths:**

1. The proposed KG-based domain adaptation for the object detection model is novel and quite interesting.
2. The result is good, comparing the prior domain adaptation methods.

**Weaknesses:**

1. The proposed method is only validated on the Detic model, which is very limited. It would be good to validate it on more LVDs and even on the open-vocabulary detectors.
2. The experiments are too naive, and can not provide a deeper understanding of the proposed method. For example, how the KG is qualitatively constructed and how it helps domain adaptation.

**Questions:**

N.A.